# OFDM Chirp Waveform Design Based on Subchirp Bandwidth Overlap and Segmented Transmitting for Low Correlation Interference in MIMO Radar

**DOI:** 10.3390/s19122696

**Published:** 2019-06-14

**Authors:** Xiang Lan, Min Zhang, Jin-Xing Li

**Affiliations:** School of Physics and Optoelectronic Engineering, Xidian University, Xi’an 710071, China; lanxiangxd@gmail.com (X.L.); jxli@stu.xidian.edu.cn (J.-X.L.)

**Keywords:** time-frequency structure, autocorrelation sidelobe interference, cross-correlation interference, orthogonal frequency-division multiplexing (OFDM) chirp waveform, multiple-input multiple-output (MIMO) radar

## Abstract

There are some special merits for the orthogonal frequency division multiplexing (OFDM) chirp waveform as multiple input multiple output (MIMO) signals. This signal has high range resolution, good Doppler tolerance, and constant modulus superiority since it exploits a full bandwidth and is based on chirp signals. The correlation sidelobe peaks level are critical for the detection requirement of MIMO radar signals, however conventional OFDM chirp signals produce high autocorrelation sidelobe peaks (ASP) and cross-correlation peaks (CP), which reduces detection performance. In this paper, we explore the structure of OFDM chirp signals’ autocorrelation function and proposed a scheme to reduce the designed signal’s ASP by a designing suitable range of subchirp bandwidth and a segmented transmit-receive mode. Next, we explore a suitable range of interval between the chirp rates of each two signals to reduce the CP. The simulation of designed signals verifies the effectiveness of the proposed methods in the reduction of ASP and CP, with the correlation performance being compared with recent relate studies. In addition, the multiple signals detection and one-dimensional range image simulation show the good detection performance of a designed signal in MIMO radar detection.

## 1. Introduction

Multiple input multiple output (MIMO) radar is the radar which can transmit multiple orthogonal signals and then receive them together to get multi-dimensional information. In order to obtain the properties of high range resolution and weak target detection, the corresponding waveform should be designed with a wider bandwidth and lower autocorrelation sidelobe peaks (ASP) than before. In addition, the cross-correlation peaks (CP) between each two waveforms need to be reduced to satisfy the orthogonality of the MIMO radar signals.

Traditionally, multiple sub-bandwidth approaches were divided to obtain the orthogonality among signals, which left limited bandwith and insufficient bandwidth for each signal. The orthogonal frequency division multiplexing (OFDM) signal was first proposed in the radar system to fully utilize the bandwidth for high range resolution [1]. The OFDM signal suggests low mutual interference between nearby radar instruments which is verified by the ambiguity function. For the advantage and widespread use of OFDM signals, numerous new techniques were generated. Under uniform circular array (UCA) and near-field conditions, the OFDM signal was combined to design a closed-form algorithm for localization [2]. A time and code OFDM algorithm was proposed to solve the large residual carrier errors existing in complex indoor environments [3]. Literature [4] designed a mutual-information OFDM waveform based on MIMO radar for low-grazing angle tracking and achieved performance improvements verified by realistic physical modeling due to adaptive OFDM waveform design. During different environments, there are many different spectrum sensing methods for OFDM signals proposed in [5,6,7,8], which improved the detection application of OFDM signals. Based on the orthogonality of it, OFDM signal was revised to complete micro-doppler estimation and the detection in frequency-selective fading channels [9,10]. In the application of passive radar signal processing, OFDM waveforms were chosen for being easily decoded to acquire a noise-free signal [11]. Moreover, the OFDM signal was used to raise the resolution, due to its high range resolution, and was investigated for range ambiguity suppression for its diversity superiority in synthetic aperture radar (SAR) images [12,13]. In addition, [14,15,16] completed the simulation and implementation of it in SAR.

Based on intrapulse radar-embedded communications, new waveforms were designed for covert and multiple target optimization, where the design criterion of the constrained maximization of the signal-to-interference ratio and constrained minimization of a suitable correlation index improve the signal detection performance [17,18]. Furthermore, it was designed to become more robust based on the polarimetric radar, considering the worst case signal-to-interference plus noise ratio [19]. Thus, in order to obtain a better Doppler tolerance and constant modulus, a new OFDM signal combined with chirp signals was proposed, called OFDM chirp signals, which had great potential in radar application. For exploiting the full bandwidth for each waveform to improve the resolution in SAR, a novel OFDM chirp waveform was raised for multiple transmitters [20]. In the study of [21], the authors designed communication-embedded OFDM chirp waveforms for delay-Doppler radar applications. The benefit of fully using the bandwidth of a OFDM chirp signal is simultaneously using it for information transmission and radar sensing. OFDM chirp signals, based on MIMO radar, was simulated in a low-grazing angle target detection for its superiority of low peak-to-average-power ratio level and its larger time-bandwidth product [22]. However, there were still some embedded problems left for OFDM chirp waveform. It has high ASP in the central region for basing chirp signal and high CP when the bandwidth is fully used. For the OFDM chirp signal, many improvement measures have been proposed. A Spread Spectrum-Coded OFDM chirp waveform was proposed in [23]. By examining the ambiguity function and correlation function, the designed waveform could stay orthogonal on the receiver and have a large time-bandwidth product for separating closely spaced targets. In addition, based on random matrix modulation, a new OFDM chirp signal with low peak–average ratios and low frequency-shift correlation peaks was designed in [24]. This was also an effective way to reduce high CP. Considering that high CP were caused by the same chirp rate of the OFDM chirp waveform, reference [25] raised various sub-chirp durations or sub-chirp bandwidths. However, the above methods only consider the case of CP’s reduction and leave high ASP unsolved, since the spectrum structure of signal remains unchanged. Moreover, we found that the method for reducing ASP may restrict the reduction of CP and thus a model should be proposed to reduce both the ASP and CP for better detection.

In this paper, a series of methods are introduced to reduce ASP and CP of conventional OFDM chirp waveforms. Firstly, we derive and analyze the autocorrelation function formula and explore a suitable range of subchirp bandwidth so as to find the lowest autocorrelation central sidelobe peaks (ACSP). To remove multiple sub-cross-correlation peaks (MSCP), which produce high peaks in ASP as subchirp bandwidth are added, we propose a transmit-receive mode of transmitting the subchirp durations one by one and superimposing each matched filtering output at the receiver during the time domain. Next, the suitable interval is designed between the chirp rates of two signals to reduce the CP. Lastly, the designed signals’ properties are evaluated by simulation of the self-ambiguity function, correlation function, and one-dimensional range image, which shows remarkable improvement in range side lobe property and orthogonality as suitable MIMO radar signals. In addition, correlation function peaks’ value is compared with other recent studies, which proves the designed signal has a better correlation performance.

The rest of the sections of this paper are organized as follows. Section 2 introduces the signal model and correlation function of the conventional OFDM chirp waveform. The autocorrelation function is explored and a new OFDM chirp waveform with lower ASP than a conventional one is proposed in Section 3. In Section 4, we explore the cross-correlation function and design a suitable range of interval between the chirp rates of two relative signals. The examples and corresponding simulation results are provided in Section 5. Finally, this paper is concluded in Section 6.

Notation: In the rest of the paper, boldface characters denote vectors. We use the upper indices to denote the type of variable and lower indices are used to denote the order. The upper indices ‘${}^{suba}$’, ‘${}^{subc}$’, ‘${}^{center}$’, and ‘${}^{\prime }$’ denote sub-autocorrelation, sub-cross-correlation, center range around main lobe, and designed one which is distinguished from the conventional one. ε(.) denotes the jump function.

## 2. Conventional OFDM Chirp Signal Model

A conventional OFDM chirp signal is made up of several subchirps, which are in different subcarrier frequencies. Each subchirp has the same subchirp bandwidth and subchirp duration for simpler modulation. In addition, each waveform has a unique code sequence of the subcarrier frequencies for orthogonality to the waveforms of the other antennas. The *n*th conventional OFDM chirp signal with *M* subchirps can be expressed as:(1)$$\begin{array}{l} s_{n}(t)=\sum_{m=0}^{M-1}s_{n}(m,t)\\ s_{n}(m,t)=(\varepsilon (t-mT_{d})-\varepsilon (t-mT_{d}-T_{d}))\cdot \exp[j2\pi (f_{nm}(t-mT_{d})+k_{n}{\left(t-mT_{d}\right)}^{2}/2)] \end{array},$$
where $s_{n}(t)$, *n* = 1, 2, …, *N* is the *n*th transmitting signal and *N* is the number of transmitting signals. $s_{n}(m,t)$ is the *m*th subchirp signal of $s_{n}(t)$. $T=MT_{d}$ denotes the total duration, $T_{d}$ is the subchirp duration, and *M* is the number of subchirp. The *t*, where 0≤t≤T, represents the time samples of the signal. In addition, ε(t)=1,t≥0 is the jump function. $B=(M-1)B_{n}^{d}+B_{n}^{l}$ represents the bandwidth of the signal, $B_{n}^{l}$ denotes the subchirp bandwidth of the *n*th transmitting signal, and $B_{n}^{d}$ denotes the minimum interval between two subcarrier frequencies of the *n*th transmitting signal. $f_{mn}$ is a subcarrier frequency, which is the starting frequency of the *m*th subchirp of the *n*th signal. $f_{nm}=C_{nm}B_{n}^{d}$, $C_{nm}$ is the subcarrier frequency code. Lastly, we define $k_{n}=B_{n}^{l}/T_{d}$ as a chirp rate of the *n*th transmitting signal since all subchirp rates are the same during an OFDM chirp signal. 

According to the signal model in Equation (1), conventional OFDM chirp waveforms with 16 subchirps are designed [24]. Without loss of generality, the conventional OFDM chirp signals $s_{1}(t)$ and $s_{2}(t)$ are chosen to explore the correlation property. We define the parameter as: B=400 M, T=8 μs, *M* = 16, *N* = 2, and $B_{n}^{l}=B/M$. The subcarrier frequency code sequences are given as: $C_{1}$ = {6, 2, 11, 5, 10, 4, 9, 7, 14, 8, 15, 16, 1, 3, 13, 12} and $C_{2}$ = {6, 10, 4, 13, 2, 7, 1, 12, 14, 9, 8, 5, 16, 15, 11, 3}. In order to do a comparison with designed signals below, we set signal $s_{1}$ with plus subchirp rates and $s_{2}$ with minus subchirp rates. Signals’ time-frequency structures diagram are shown in Figure 1. 

By fully using the bandwidth based on the subchirp signal, the conventional OFDM chirp signals have a high resolution with no range-Doppler coupling. To better evaluate the property of conventional OFDM chirp signals, the correlation function [25] of the two waveforms are defined as: (2)$$c_{pq}(\tau )=\left\{ \begin{array}{ll} \int_{\tau }^{T}s_{p}(t)s_{q}{}^{*}(t-\tau )dt, & 0<\tau <T\\ \int_{0}^{T+\tau }s_{p}(t)s_{q}{}^{*}(t-\tau ), & -T<\tau \leq 0 \end{array}\right.,$$
where τ is the time delay, $c_{pq}(\tau )$ is the cross-correlation function under p≠q, and it represents autocorrelation function after p=q. The autocorrelation function of $s_{1}(t)$ and cross-correlation function between $s_{1}(t)$ and $s_{2}(t)$ calculated by Equation (2) is obtained in Figure 2.

As show in Figure 2a, it can be found that a conventional OFDM signal’s ASP is up to −13.4 dB. In addition, the ASP appears in the central region near the main lobe (called ACSP), while the autocorrelation side peaks in the edge region are low enough. On the other hand, Figure 2b shows the signal’s CP value as −25 dB with many high cross-correlation peaks in it.

These numerical results show that a high correlation influence exists in conventional OFDM signals, which will reduce its detection properties. As the signals’ number add, such as 4 signals in the Section 5 simulation, the cross-correlation influence will be more serious. Moreover, in the one-dimensional range image, multiple point targets echo’s correlation sidelobes will overlap and become higher, which may cover up the weak targets and cause false detection. Thus, it is necessary to reduce the ASP and CP of a conventional OFDM signal and evaluate the multiple signals detection performance and one-dimensional imaging detection level of the designed signal. In addition, some recent study results should be compared with the simulation results of the designed signals. 

## 3. Subchirp Bandwidth and Transmitting Structure Design for Reducing ASP

In this section, we begin with the derivation and analysis of the autocorrelation function of a conventional OFDM signal. Next, we separate the function into several parts and evaluate each part’s influence on the ASP. Lastly, we propose the designed signal structure and show corresponding autocorrelation performance evaluated by a transmit-receive structure diagram, time-frequency structure figure, and ASP curve. 

Under the situation of *p* = *q* = *n*, Equation (2) is the auto-correlation function of the signals. Since the function is probably symmetric by τ=0, which is sufficient to study the peak value in 0≤τ<T, Equation (2) can be written as:(3)$$c_{nn}(\tau )=\int_{\tau }^{T}s_{n}(t)s_{n}{}^{*}(t-\tau )dt,\;\;\;\;\begin{array}{c} 0<\tau <T \end{array},$$
We assume $l\cdot T_{d}\leq \tau <(l+1)T_{d}$, where *l* is an integer with l≥0, and set the signal parameters as the same as in Figure 1. Based on Equations (1) and (3), the signal’s autocorrelation function structure diagram is obtained in Figure 3. Next, based on the different integral function as in Figure 3, the region is divided into 2*M* − 2*l* − 1 regions and Equation (3) can be expanded as: (4)$$\begin{array}{ll} c_{nn}(\tau )= & \sum_{m=0}^{M-l-1}\int_{\tau +mT_{d}}^{(l+1+m)T_{d}}s_{n}(m,t-lT_{d}-mT_{d})s_{n}^{*}(m,t-\tau -mT_{d})dt\\ & +\sum_{m=0}^{M-l-2}\int_{(l+1+m)T_{d}}^{\tau +(1+m)T_{d}}s_{n}(m,t-(l+1)T_{d}-mT_{d})s_{n}^{*}(m,t-\tau -mT_{d})dt\\ & =c_{1}(\tau )+c_{2}(\tau ) \end{array},$$
where $c_{1}(\tau )$ and $c_{2}(\tau )$ are the first and second part of $c_{nn}(\tau )$. 

In Figure 3, the conjugate multiplication of the two signals’ overlap with interval of τ is the corresponding value of the autocorrelation function (AF). Moreover, we define the part of AF composed by two same subchirps as the multiple sub-autocorrelation function (MSAF), and the other one that is composed of two different subchirps as the multiple sub-cross-correlation function (MSCF). Under $t=\tau_{1}$ of Figure 3, AF is made up of the MSAF and MSCF. The part in $\tau_{1}+mT_{d}\sim T_{d}+mT_{d}$ of AF is MSAF, which is the first part of Equation (4) and can be expressed as: (5)$$c_{nn}^{suba}(\tau_{1})=c_{1}(\tau_{1}),\;0\leq \tau_{1}<T_{d},$$

The one in $T_{d}+mT_{d}\sim \tau_{1}+(m+1)T_{d}$ is the MSCF which is the second part of Equation (4) and can be expressed as: (6)$$c_{nn}^{subc}(\tau_{1})=c_{2}(\tau_{1}),\;0\leq \tau_{1}<T_{d},$$
Under $t=\tau_{2}$, the AF is made up of the MSCF only and can be expressed as: (7)$$c_{nn}^{subc}(\tau_{2})=c_{1}(\tau_{2})+c_{2}(\tau_{2})\;,\tau_{2}\geq T_{d},$$

In order to reduce the ASP of the signals, the auto-correlation curve structure can be adjusted by changing its spectral structure. The subchirp bandwidth $B_{n}^{l}$ has been changed to find a suitably low ASP value. We define:(8)$$p_{1}=\max(c_{nn}(\tau ),0<\tau \leq T),$$
(9)$$p_{2}=\max(c_{nn}^{suba}(\tau ,0<\tau \leq T)),$$
(10)$$p_{3}=\max(c_{nn}^{subc}(\tau ,0<\tau \leq T)).$$
where $p_{1}$ is the autocorrelation sidelobe peak (ASP), $p_{2}$ is the multiple sub-autocorrelation sidelobe peak (MSASP), and $p_{3}$ is the multiple sub-cross-correlation peak (MSCP), where $p_{1}=\max(p_{2},p_{3})$. According to Equations (4)–(7), their curves change with $B_{n}^{l}$ as shown in Figure 4. The MSASP has two minimal values when $B_{n}^{l}/(B/16)$ is near 6.97 or 9.62, and the MSCP and ASP have high value in these points. 

According to Figure 2a, the high ASP appears in the central region near the main lobe. We explore the 1/200$T_{d}$ region centered on the main lobe where τ is smaller than 1/200$T_{d}$ and the second part of Equation (4) is about −46 dB in value of the first one. Thus, the AF in the central region can be obtained without the second part in Equation (4) as: (11)$$c_{nn}^{center}(\tau )\approx c_{1}(l=0)\;0\leq \tau <<T_{d},$$

Comparing Equation (11) with Equation (5), it shows $co_{center}^{nn}(\tau )\approx co_{suba}^{nn}(\tau )$ during $0\leq \tau <<T_{d}$. Thus, as the MSASP curve obtained in Figure 4 shows, the central ASP can reduce to about −30 dB when $B_{n}^{l}/(B/16)$ takes a suitable value. However, the MSCP curve will appear at high peaks when $B_{n}^{l}/(B/16)>1$ which will cause new high ASP out of the central region of the main lobe. 

Next, to find the reason for the high MSCP according to Equation (6), Equation (7) is qualitatively analyzed. We expand the multiple sub-cross-correlation function $c_{nn}^{subc}(\tau )$ during $\tau >T_{d}$ as:(12)$$\begin{array}{l} \begin{array}{ll} c_{nn}^{subc}(\tau )= & \sum_{m=0}^{M-l-1}\int_{\tau +mT_{d}}^{(l+1+m)T_{d}}s_{n}(m,t-lT_{d}-mT_{d})s_{n}^{*}(m,t-\tau -mT_{d})dt\\ & +\sum_{m=0}^{M-l-2}\int_{(l+1+m)T_{d}}^{\tau +(1+m)T_{d}}s_{n}(m,t-(l+1)T_{d}-mT_{d})s_{n}^{*}(m,t-\tau -mT_{d})dt\\ & =\sum_{m=0}^{M-l-1}\int_{\tau +mT_{d}}^{(l+1+m)T_{d}}\exp(2\pi j((f_{n(l+m)}-f_{nm}+k(\tau -lT_{d}))t)\cdot \exp(2\pi j(\varphi_{0}))dt\\ & +\sum_{m=0}^{M-l-2}\int_{\tau +mT_{d}}^{(l+1+m)T_{d}}\exp(2\pi j((f_{n(l+m+1)}-f_{nm}+k(\tau -(l+1)T_{d}))t\cdot \exp(2\pi j(\varphi_{1}))dt \end{array}\\ \tau >T_{d}\\ \varphi_{0}=-f_{n(l+m)}((l+m)T_{d})+f_{nm}(mT_{d}+\tau )+1/2k({\left((l+m)T_{d}\right)}^{2}-{\left(mT_{d}+\tau \right)}^{2})\\ \varphi_{1}=-f_{n(l+m+1)}((l+m+1)T_{d})+f_{nm}(mT_{d}+\tau )+1/2k({\left((l+m+1)T_{d}\right)}^{2}-{\left(mT_{d}+\tau \right)}^{2}) \end{array}.$$
where $\left|f_{n(m+l)}-f_{nm}\right|\geq B_{n}^{d}$. When $B_{n}^{l}\leq B_{n}^{d}$, $\left|k(\tau -(l+1)T_{d})\right|\neq \left|f_{n(l+m+1)}-f_{nm}\right|$ and $\left|k(\tau -lT_{d})\right|\neq \left|f_{n(l+m)}-f_{nm}\right|$, MSCF will not produce high peaks. However, when $B_{n}^{l}>B_{n}^{d}$, $\left|k(\tau -(l+1)T_{d})\right|=\left|f_{n(l+m+1)}-f_{nm}\right|$, or $\left|k(\tau -lT_{d})\right|=\left|f_{n(l+m)}-f_{nm}\right|$ MSCP will produce high peaks. In addition, the conclusion is the same for $c_{nn}^{subc}(\tau )$ when $0\leq \tau \leq T_{d}$.

Thus, it can be concluded that a suitable value of $B_{n}^{l}$ can reduce the ASP in the central range of the main lobe in AF. However, the MSCP will keep a high value as the $B_{n}^{l}$ increases to more than $B_{n}^{d}$, which cause new edge high ASP values. A designed transmit-receive structure needs to be proposed to remove the MSCF part from the AF.

The autocorrelation function of *m*th subchirp signal can be written as:(13)$$\begin{array}{l} c_{nn}^{suba}(m,\tau )=\int_{\tau +mT_{d}}^{(1+m)T_{d}}s_{n}(m,t)\cdot s_{n}{}^{*}(m,t-\tau )dt\\ \;0\leq \tau <T_{d} \end{array},$$
And Equation (3) can be expanded and rewritten as:(14)$$c_{nn}^{suba}(\tau )=\sum_{m=0}^{M-1}c_{nn}^{suba}(m,\tau )\;\;\begin{array}{c} 0\leq \tau <T_{d} \end{array},$$
where MSAF is the sum of *M* subchirp autocorrelation functions. Inspired by Equation (14), designed signals can be designed for removing the MSCF part by a new transmit-receive mode based on the OFDM chirp signal as shown in Figure 5 where it is compared with a conventional signal.

(1) In Figure 5a, all subchirps are continuously transmitted in a pulse with each subchirp duration $T_{d}=T/M$ in the transmission of the conventional OFDM signal. However, in the transmission of designed signal, each subchirp duration is $T_{d}=T$ and is transmitted in different pulses according to the code order. Where $T_{p}$ is the pulse duration and is set to *M* = 2 to simplify the autocorrelation structure;

(2) In Figure 5b, the same matched filter output is obtained in a pulse in conventional OFDM signal processing. However, in the designed signal processing, *M* different matched filter outputs are produced among *M* pulses of time domain in the order of the transmitting subchirps;

(3) After matched filtering, *M* pulse durations outputs are accumulated at the receive. Both signals will improve the SNR and the designed one can also achieve a low ASP. 

As mentioned above, the designed signal can be formulated as: (15)$$\begin{array}{l} {s^{\prime }}_{n}(t)=\sum_{m=0}^{M-1}{s^{\prime }}_{n}(m,t)\\ \begin{array}{ll} {s^{\prime }}_{n}(m,t)= & \left(\varepsilon (t-mT_{p})-\varepsilon (t-mT_{p}-T_{p})\right)\\ & \cdot \exp[j2\pi (f_{nm}(t-mT_{p})+k_{n}{\left(t-mT_{p}\right)}^{2}/2)] \end{array} \end{array},$$
where ${s^{\prime }}_{n}(t)$, *n* = 1, 2, …, *N* is the *n*th designed OFDM chirp transmitting signals. The pulse duration is $T_{p}$ and the subchirp duration is T, where $T<T_{p}$. The range of *t* is $0\leq t\leq M\cdot T_{p}$. In addition, $B_{n}^{l}$ will be chosen during the suitable range to reduce the ASP. 

To explore the autocorrelation property of the designed signals, we set the parameters as: B=400 M, T=8 μs, *M* = 16, *N* = 1, and $B_{n}^{l}=6.97B/M$ according to the conclusion in Figure 4. In addition, the subchirp carrier frequency code sequences are the same as the conventional OFDM chirp signal $s_{1}$ in Figure 1. The time-frequency structure of the designed waveform is plotted in Figure 6. Where Figure 6a is the first part of Figure 6b during $0\sim T_{p}$, which shows each subchirp duration of the designed waveform exists in a unique pulse. Moreover, Figure 6b plots the whole time-frequency structure of the designed signal during $0\sim MT_{p}$. It shows each subchirp rate is the same and overlap in the frequency dimension since $B_{n}^{l}=6.97B/M$.

Compared with Equation (2), the correlation function of designed OFDM chirp signals can be defined as ${c^{\prime }}_{pq}(\tau )$. Since the subchirp signals of the designed signals are transmitted according to the pulse one by one and the *M* outputs are superimposed at time domain without delay, ${c^{\prime }}_{pq}(\tau )$ can be written as: (16)$$\begin{array}{l} {c^{\prime }}_{pq}(\tau )=\sum_{m=0}^{M-1}{c^{\prime }}_{pq}(m,\tau )\\ {c^{\prime }}_{pq}(m,\tau )=\left\{ \begin{array}{l} \int_{\tau +mT_{p}}^{T+mT_{p}}{s^{\prime }}_{p}(m,t){s^{\prime }}_{q}{}^{*}(m,t-\tau )dt\;\;0\leq \tau \leq T\\ \int_{mT_{p}}^{T+\tau +mT_{p}}{s^{\prime }}_{p}(m,t){s^{\prime }}_{q}{}^{*}(m,t-\tau )dt\;-T\leq \tau <0 \end{array}\right. \end{array},$$
the autocorrelation function of the designed signals can be expressed as:(17)$$c{{}^{\prime }}_{nn}(\tau )=\sum_{m=0}^{M-1}c{{}^{\prime }}_{nn}(m,\tau ),$$
and its ASP can be defined as:(18)$$p_{4}=\max(c{{}^{\prime }}_{nn}(\tau ),0<\tau \leq T),$$
According to Equations (15)–(18), the designed signal’s ASP curve with $B_{n}^{l}$ is plotted in Figure 7 and is compared with the MSASP conventional signal. 

In Figure 7, since the ASP of the designed signal is equal to the MSASP of the conventional one, the designed signals’ ASP can be suppressed effectively when taking a specific $B_{n}^{l}$ value. The suitable range of $B_{n}^{l}$ when designed signal’s ASP is under −30 dB can be obtained, which is shown in Table 1.

## 4. Chirp Rates Interval Design for Reducing CP

Considering the correlation function of the designed signal, Equation (16) can be written under p≠q as:(19)$$\begin{array}{l} c{{}^{\prime }}_{pq}(\tau )=\sum_{m=0}^{M-1}c{{}^{\prime }}_{pq}(m,\tau )\\ c{{}^{\prime }}_{pq}(m,\tau )=\left\{ \begin{array}{l} \int_{\tau +mT_{p}}^{T+mT_{p}}s{{}^{\prime }}_{n}(m,t)s{{}^{\prime }}_{n}{}^{*}(m,t-\tau )dt\;\;0<\tau <T\\ \int_{mT_{p}}^{T+\tau +mT_{p}}s{{}^{\prime }}_{n}(m,t)s{{}^{\prime }}_{n}{}^{*}(m,t-\tau )dt\;-T<\tau <0 \end{array}\right. \end{array},$$
where $c{{}^{\prime }}_{pq}(\tau )$ is the cross-correlation function between $s{{}^{\prime }}_{p}(t)$ and $s{{}^{\prime }}_{q}(t)$, and $c{{}^{\prime }}_{pq}(m,\tau )$ is the *m*th pulse of it. We expand $c{{}^{\prime }}_{pq}(\tau )$ as:(20)$$\begin{array}{l} c{{}^{\prime }}_{pq}(\tau )=\sum_{m=0}^{M-1}\int_{\tau +mT_{p}}^{T+mT_{p}}\exp(2\pi j((f_{pm}-f_{qm}+k_{q}\tau )t+(k_{p}-k_{q})(1/2t^{2}+mT_{p}t))dt\cdot \exp(2\pi j(\varphi_{0}))\\ \varphi_{0}=-f_{pm}(mT_{p})+f_{qm}(mT_{p}+\tau )\;+1/2(k_{p}{\left(mT_{p}\right)}^{2}-k_{q}{\left(mT_{p}+\tau \right)}^{2})\;\;\;\;\;\;\;\;\;\;\;\;\;\;\;\;\;\;\;\;\;\;\;0\leq \tau \leq T \end{array},$$
(21)$$\begin{array}{l} c{{}^{\prime }}_{pq}(\tau )=\sum_{m=0}^{M-1}\int_{mT_{p}}^{T+\tau +mT_{p}}\exp(2\pi j((f_{pm}-f_{qm}+k_{q}\tau )t+(k_{p}-k_{q})(1/2t^{2}+mT_{p}t))dt\cdot \exp(2\pi j(\varphi_{0}))\\ \varphi_{0}=-f_{pm}(mT_{p})+f_{qm}(mT_{p}+\tau )+1/2(k_{p}{\left(mT_{p}\right)}^{2}-k_{q}{\left(mT_{p}+\tau \right)}^{2})\;\;\;\;\;\;\;\;\;\;\;\;\;\;\;\;\;\;\;\;\;\;\;-T\leq \tau <0 \end{array},$$
where owing to $0\leq \left|k\tau \right|\leq B_{l}$ and $-B\leq f_{pm}-f_{qm}\leq B$, $(f_{pm}-f_{qm}+k_{q}\tau )=0$ will happen during −T≤τ≤T. Furthermore, under $k_{p}=k_{q}$, if $(f_{pm}-f_{qm}+k_{q}\tau )=0$, function $co{{}^{\prime }}_{pq}(m,\tau )$ would have a maximum value, which causes high CP. Thus, when the time-frequency structure in Section 2 is unchanged, $k_{p}\neq k_{q}$ should be established to avoid high CP. We define Δk as: (22)$$\Delta k=k_{p}-k_{q},$$
and define the CP between $s{{}^{\prime }}_{p}(t)$ and $s{{}^{\prime }}_{q}(t)$ as:(23)$$p_{5}=\max(co{{}^{\prime }}_{pq}(\tau ),-T\leq \tau <T).$$

Next, the relation between the CP and Δk is explored. We set $k_{q}=8(\frac{B}{MT})$, and keep $k_{p}$ changing during $0\leq k_{p}\leq 16(\frac{B}{MT})$. Moreover, the subcarrier frequency code sequences are set the same as the conventional OFDM chirp signal in Figure 1. Lastly, the CP curve of the designed signals $s{{}^{\prime }}_{p}(t)$ and $s{{}^{\prime }}_{q}(t)$ with chirp rate difference Δk is plotted basing Equations (20)–(23) in Figure 8. 

In Figure 8, the CP shows a remarkable reduction with Δk’s increase. The range of interval between the chirp rates of two designed signals for reducing their CP under −30 dB is shown in Table 2. When Δk=0, the CP is −19.5 dB, which reduces to −30dB while Δk is taken as −0.165 $\left(\frac{B}{MT}\right)$. Thus, low CP can be obtained with a suitable interval between the chirp rates of two designed signals.

## 5. Design Examples and Simulation Results

In this section, we give some designed examples and corresponding simulation results to evaluate the effectiveness of the proposed OFDM chirp waveform methods. 

To explore the designed signals’ Doppler performance, Figure 9a–f gives the self-ambiguity function response of a conventional OFDM signal and designed signal with its $B_{}^{l}=6.8/16B$, and parameters at B=40 M and T=8 μs. As shown in Figure 9b,e, the designed signal has the same doppler resolution performance as a conventional OFDM chirp signal. Furthermore, in Figure 9c,f, the designed signal has lower range sidelobes than conventional ones.

In order to evaluate the time-frequency structure and correlation performance of the designed signals, four signals $s_{1}{}^{\prime },\;s_{2}{}^{\prime },\;s_{3}{}^{\prime },\;s_{4}{}^{\prime }$ are simulated. According to the conclusion in Table 1, there are two ranges for the $B^{l}$ to obtain low ASP. We designed two groups of four signals with each two taking the $B^{l}$ value during each range respectively for the simulation. In addition, as shown in the conclusion in Table 2, the interval of the chirp rates between the two simulation signals will be more than $-0.165(\frac{B}{MT})$. The designed parameters are shown in Table 3. 

After subchirps sequences coding and subchirp rate plus and minus (PM) coding for reducing the designed signals’ CP, the time-frequency structures of the designed four OFDM chirp signals are plotted in Figure 10a–d, where $s_{1}{}^{\prime }$ and $s_{2}{}^{\prime }$ are coded as the sequences of $s_{1}$ and $s_{2}$ for comparison. In Figure 10, each subchirp duration is *T*, and in a signal, each subchirp bandwidth is $B_{n}^{l}{}^{\prime }$, where *n* is 1, 2, 3, 4. Since the PM coding will not influence the ASP of the designed signals for each subchirp being transmitted separately, we set signals $s_{1}{}^{\prime }$ and $s_{3}{}^{\prime }$ with all plus subchirp rates and signals $s_{2}{}^{\prime }$ and $s_{4}{}^{\prime }$ with all minus subchirp rates to further reduce the CP. 

Figure 11 plots the correlation functions of signals $s_{1}{}^{\prime }$ and $s_{2}{}^{\prime }$. Comparing the results in Figure 11a with that of Figure 2a shows that the autocorrelation of conventional OFDM waveforms and designed waveforms are different. Figure 2a shows high sidelobes near the main lobe, which also influences the resolution among multiple close targets or in continuous targets and easily causes false detection. The reason is that inappropriate subchirp bandwidth causes high peaks in multiple-subchirp autocorrelation function. Since the suitable subchirp bandwidths are taken and designed transmitting mode is adopted which removes the MSCF from AF, the designed waveforms’ ASP obtains an effective suppression near the main lobe, as obtained in Figure 11.

By comparing the results in Figure 11b with that of Figure 2b, it shows that the cross-correlation functions of the conventional waveforms and designed waveforms are diverse. The cross-correlation functions of the conventional waveforms in Figure 2b have some high grating sidelobes, which are easily judged as weak targets, especially when multiple sidelobes of several close targets overlap and cause false detection. The reason for this is that subchirps with the same carrier frequency and the same chirp rate of two signals simultaneously exist in the same subchirp durations. Since the chirp rates of two signals have kept enough interval as the proposed methods, the sidelobes in the cross-correlation function of the designed waveforms reduce dramatically in Figure 11b, and the sidelobe levels are more stable than that of the conventional waveforms. 

For an evaluation of the resolution property, the autocorrelation curves comparison has been made in Figure 12. Since there exists a difference among the frequency spectrums of the three signals, they have different autocorrelation function structures. The nonlinear frequency modulation (NLFM) signal has widest main lobe, 4.05 ns, and the lowest sidelobes, −52.7 dB, and the conventional OFDM signal has the narrowest main lobe, 2.16 ns, but the highest sidelobes, −13.4 dB. The designed signal takes the middle properties with the main lobe width, 3 ns, and sidelobe height, −30.2 dB, which obtains a balance between resolution and detection. Therefore, the resolution property of the designed signal reduces for the improvement of detection, which is 0.52 m compared with 0.375 m for the conventional one.

Next, for further comparisons of detection performance, the ASP and CP values of conventional OFDM (COFDM) signal, Li’s OFDM signal [25], piecewise nonlinear frequency modulation OFDM (PNLFM-OFDM) signal [26], and the designed segmented transmitting OFDM (STOFDM) signal are listed in Table 4. Where the STOFDM signal is superior with the lowest sidelobes in both ASP and CP.

To better evaluate the multiple signals detection performance, different combinations of four designed signals are simulated with their ASP and CP values listed in Table 5. The autocorrelation of signal $s_{1}{}^{\prime }+s_{2}{}^{\prime }$ and cross-correlation between signal $s_{1}{}^{\prime }+s_{2}{}^{\prime }$ and $s_{3}{}^{\prime }+s_{4}{}^{\prime }$ are shown in Figure 13. In addition, the autocorrelation of signal $s_{1}{}^{\prime }$ and cross-correlation between signal $s_{1}{}^{\prime }$ and $s_{2}{}^{\prime }+s_{3}{}^{\prime }+s_{4}{}^{\prime }$ are shown in Figure 14. The average correlation sidelobes among four designed signals $s_{1}{}^{\prime }-s_{4}{}^{\prime }$ are stable and all lower than −30 dB since the number of transmitting signals adds to 4, which proves the good sidelobe properties of STOFDM signals as MIMO radar signals.

Finally, we simulate the proposed OFDM chirp waveforms in a MIMO radar one-dimensional range imaging application. Without loss of generality, we consider a four-antenna MIMO radar using the waveforms $s_{1}{}^{\prime }-s_{4}{}^{\prime }$ illustrated in Figure 10. Other parameters are assumed as: B=100 M, T=20 μs. In a one-dimensional range image, we decomposed a ship target, which is 172 m long and 16m wide, into 152 point targets. Figure 15a shows the one-dimensional radar cross section (RCS) range images and Figure 15b shows the one of four matched filters output of $s_{1}{}^{\prime }$. For a comparison, the imaging result for a conventional single input single output (SISO) radar using a conventional OFDM chirp signal with equal system parameters is also shown in Figure 15c. Compared with Figure 15a, there are some high amplitude false target points in Figure 15c, for the high range peak sidelobe ratios (RPSLR) of conventional chirp signal. But in Figure 15b, those false targets are suppressed and real target points are compatible with Figure 15 even after adding cross-correlation interferences with $s_{2}{}^{\prime }-s_{4}{}^{\prime }$. The RPSLRs of each signal are shown in Table 6.

## 6. Conclusions

The OFDM chirp waveform is widely applied in MIMO radar because of its high range resolution and good Doppler tolerance. However, some high lobes exist in the signals’ cross-correlation function for multiplexing of bandwidth and since the signal is based on a chirp signal, high sidelobes also exist in its autocorrelation function. These two imperfections may influence weak target detection. To solve this problem, the proposed method included two design steps: Signal autocorrelation construct design and orthogonality design between signals. A range of subchirp bandwidths and a corresponding transmit-receive mode were designed to reduce ASP and a range of suitable intervals between the chirp rates of two relative signals were proposed to reduce CP. The self-ambiguity functions and correlation function verified the good Doppler tolerance, low ASP, and CP properties as well as a little main lobe broadening of the designed signal. Moreover, the multiple signals detection and one-dimensional range image simulation proved the improvement of multiple signals detection performance in MIMO radar especially for weak targets detection and low range peak sidelobe superiority of designed signals.

In future work, we intend to further optimize the multiple signals detection model and reduce the mutual impact among transmitted signals. On the other hand, the targets RCS property will be combined to analyze the signal detection performance in specific changes and targets. Thus, the application of designed signals can be improved.

## Figures and Tables

**Figure 1 sensors-19-02696-f001:**
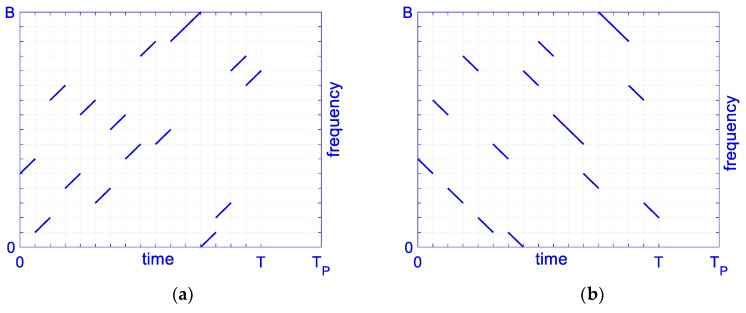
Time-frequency structure of conventional orthogonal frequency division multiplexing (OFDM) chirp waveform, *M* = 16. (**a**) $s_{1}(t)$; (**b**) $s_{2}(t)$.

**Figure 2 sensors-19-02696-f002:**
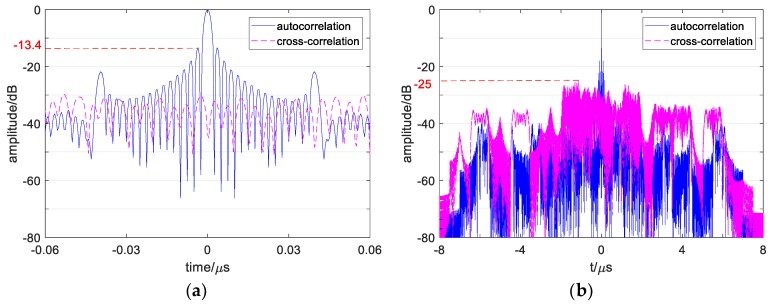
Correlation curve of conventional OFDM chirp signals. (**a**) Correlation curve of $s_{p}$ and $s_{q}$ during −0.06 μs~0.06 μs. (**b**) Correlation curve of $s_{p}$ and $s_{q}$ during −8 μs~8 μs.

**Figure 3 sensors-19-02696-f003:**
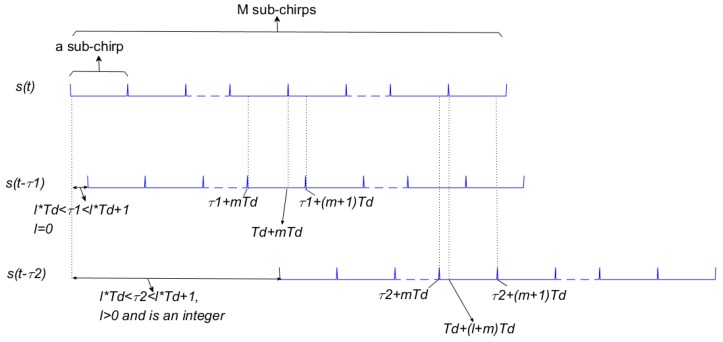
Autocorrelation function a structure diagram.

**Figure 4 sensors-19-02696-f004:**
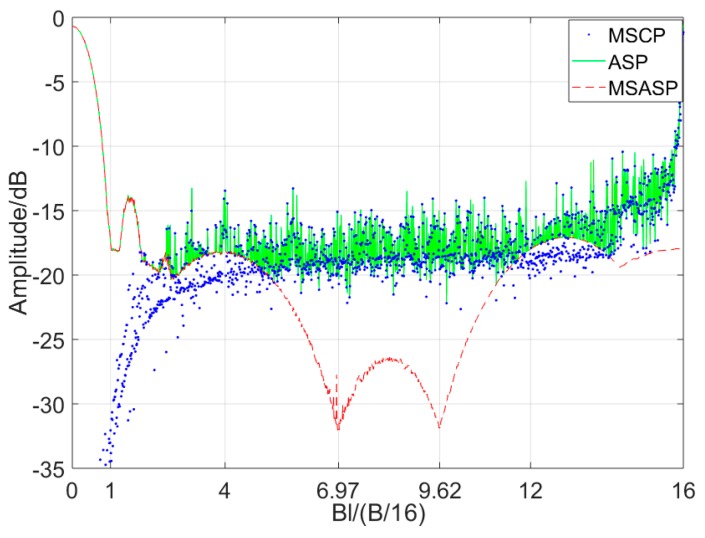
Autocorrelation sidelobe peak (ASP), multiple sub-cross-correlation peak (MSCP), and multiple sub-autocorrelation sidelobe peak (MSASP) curve of conventional OFDM chirp with $B_{n}^{l}/B/16$.

**Figure 5 sensors-19-02696-f005:**
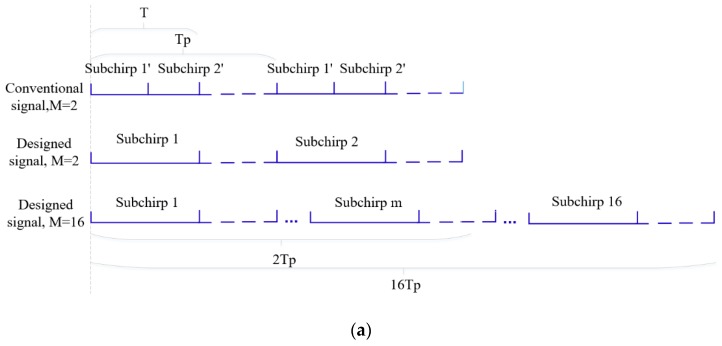
The transmitting and processing structure of designed signals: (**a**) Transmitting structures of two signals; (**b**) matched filtering output structures of two signals.

**Figure 6 sensors-19-02696-f006:**
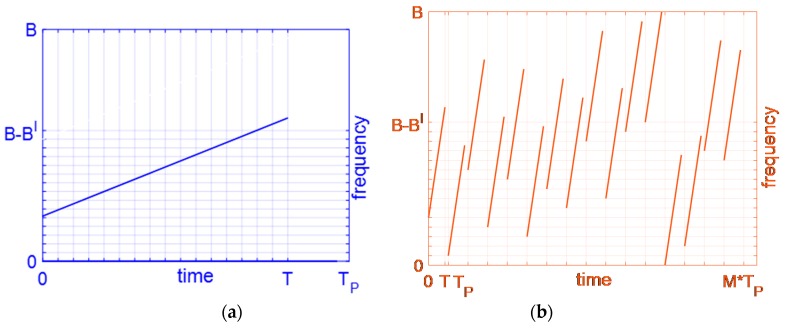
Time-frequency structure of designed OFDM chirp waveform, *M* = 16. (**a**) During a pulse and $B_{n}^{l}/B$ = 6.97/16; (**b**) During 16 pulses and $B_{n}^{l}/B$ = 6.97/16.

**Figure 7 sensors-19-02696-f007:**
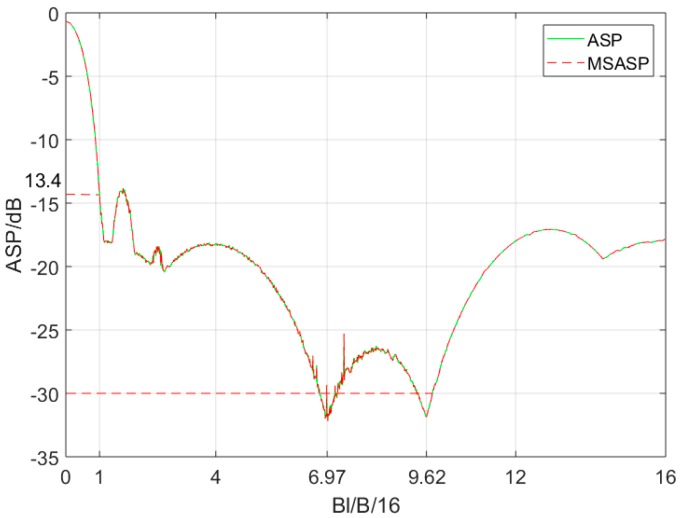
ASP curve of designed OFDM chirp and MSASP curve of conventional OFDM chirp with $B_{n}^{l}/B/16$.

**Figure 8 sensors-19-02696-f008:**
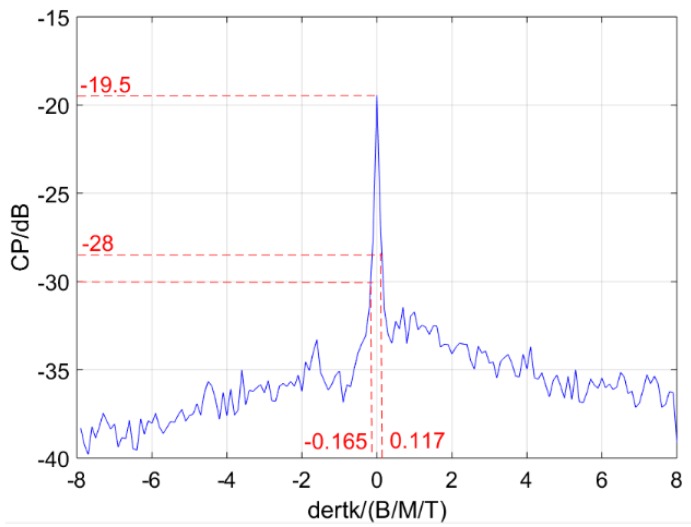
Cross-correlation peaks (CP) curve with Δk when $k_{q}=8(\frac{B}{MT})$.

**Figure 9 sensors-19-02696-f009:**
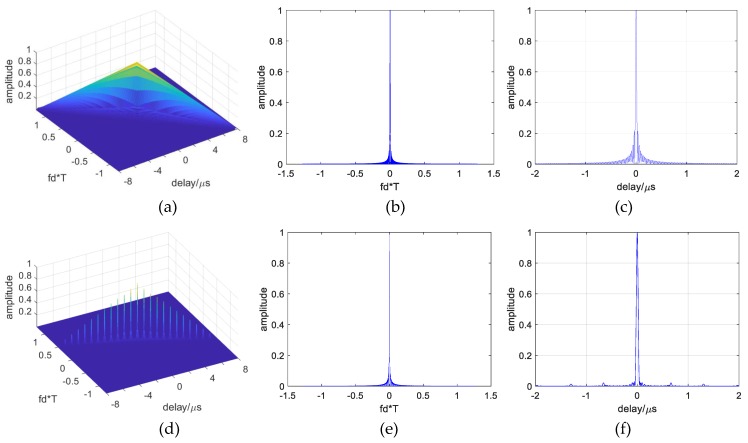
Self-ambiguity functions and their zero-delay and zero-Doppler “cuts”. (**a**) Conventional OFDM waveform; (**b**) zero-delay cut; (**c**) zero-Doppler cut; (**d**) designed OFDM chirp waveforms; (**e**) zero-delay cut; (**f**) zero-Doppler cut.

**Figure 10 sensors-19-02696-f010:**
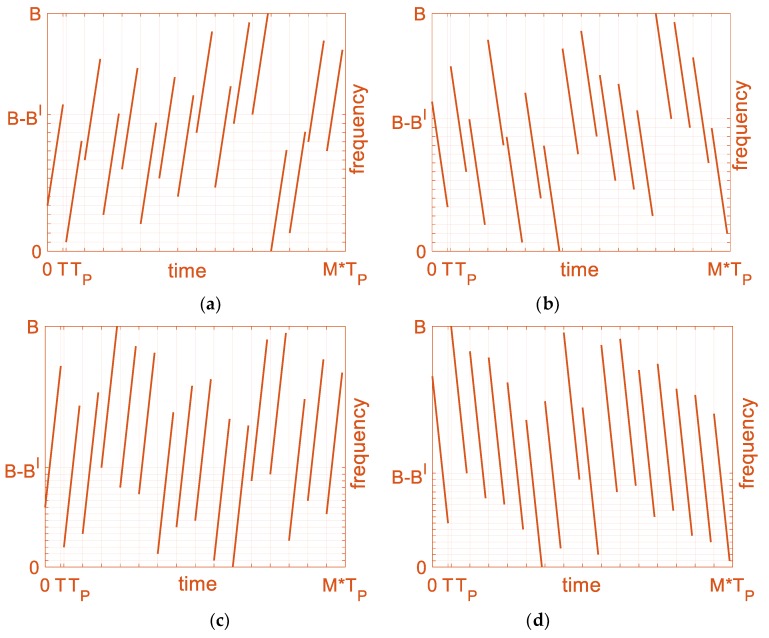
Time-frequency structure of four simulated, designed OFDM chirp signals, *M* = 16. (**a**) $s_{1}{}^{\prime }$, $B_{1}^{l}{}^{\prime }$ = 6.8/16*B*; (**b**) $s_{2}{}^{\prime }$, $B_{2}^{l}{}^{\prime }$ = 7.09/16*B*; (**c**) $s_{3}{}^{\prime }$, $B_{3}^{l}{}^{\prime }$ = 9.4/16*B*; (**d**) $s_{4}{}^{\prime }$, $B_{4}^{l}{}^{\prime }$ = 9.77/16*B*.

**Figure 11 sensors-19-02696-f011:**
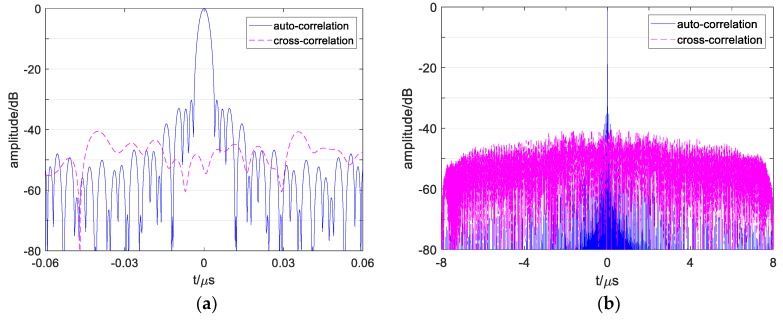
Correlation curve of designed OFDM chirp signals $s_{1}{}^{\prime }$ and $s_{2}{}^{\prime }$. (**a**) Correlation curve of $s_{1}{}^{\prime }$ and $s_{2}{}^{\prime }$ during −0.06 μs<t<0.06 μs. (**b**) Correlation curve of $s_{1}{}^{\prime }$ and $s_{2}{}^{\prime }$ during −8 μs<t<8 μs.

**Figure 12 sensors-19-02696-f012:**
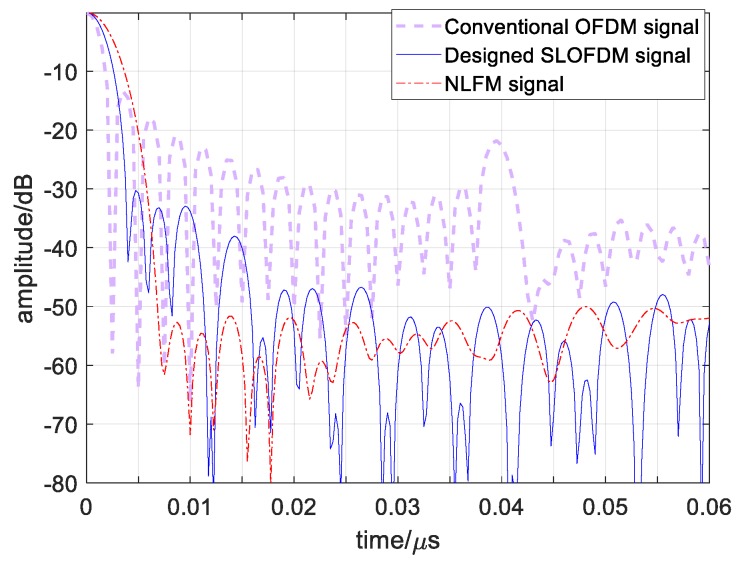
Autocorrelation curve comparison among designed OFDM chirp signals $s_{1}{}^{\prime }$, conventional signal $s_{1}$, and nonlinear frequency modulation (NLFM) signal during 0<t<0.06 μs.

**Figure 13 sensors-19-02696-f013:**
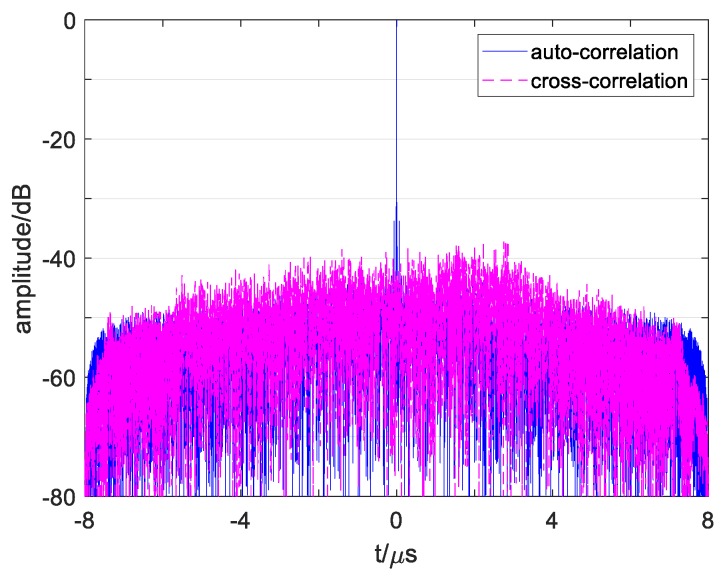
Correlation curve of designed OFDM chirp signals $s_{1}{}^{\prime }$ + $s_{2}{}^{\prime }$ and $s_{3}{}^{\prime }+s_{4}{}^{\prime }$ during −8 μs<t< μs.

**Figure 14 sensors-19-02696-f014:**
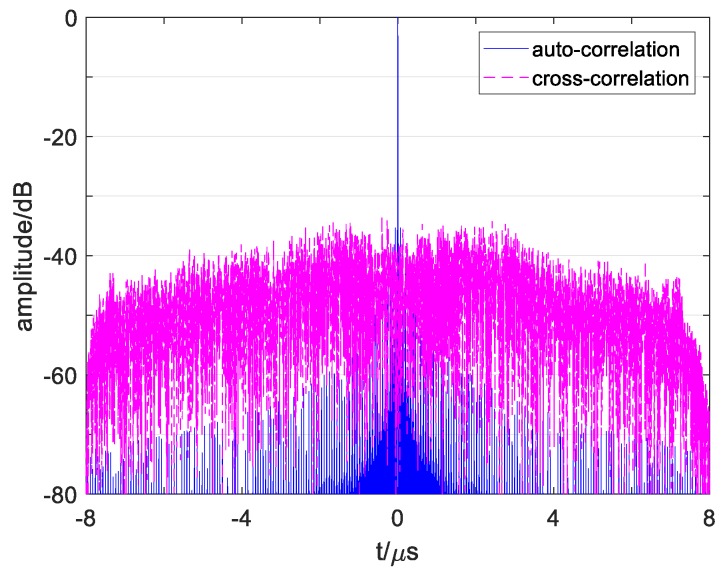
Correlation curve of designed OFDM chirp signals $s_{1}{}^{\prime }$ and $s_{2}{}^{\prime }+s_{3}{}^{\prime }+s_{4}{}^{\prime }$ during −8 μs<t<8 μs.

**Figure 15 sensors-19-02696-f015:**
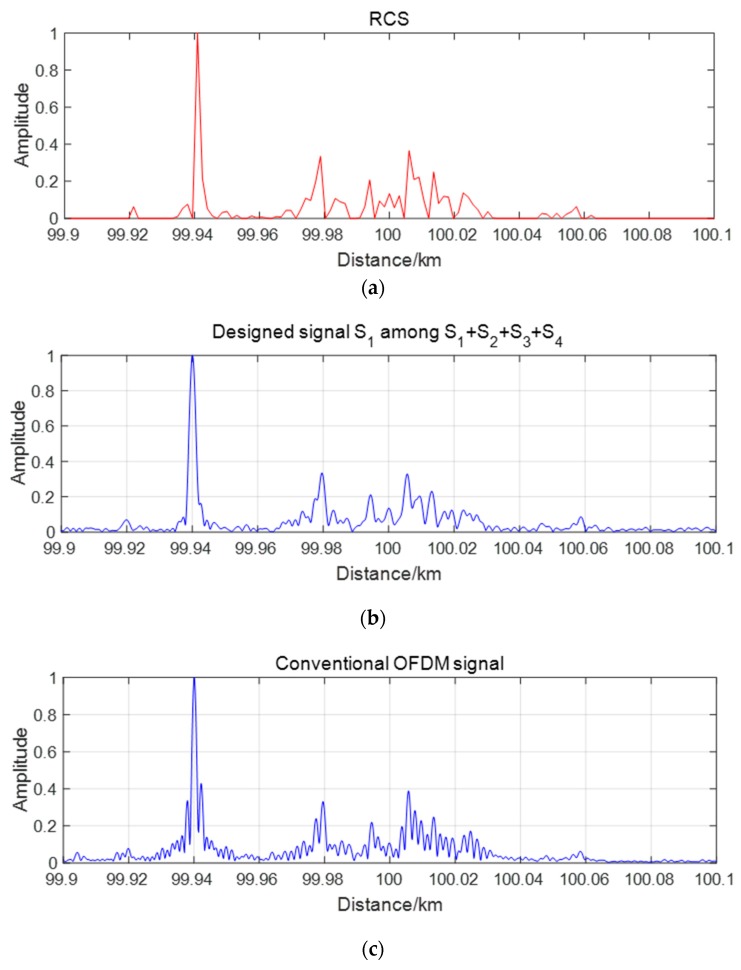
One-dimensional range image. (**a**) One-dimensional radar cross section (RCS) range images; (**b**) Designed signal $s_{1}{}^{\prime }$ during total signal $s_{1}{}^{\prime }+s_{2}{}^{\prime }+s_{3}{}^{\prime }+s_{4}{}^{\prime }$ in MIMO; (**c**) Conventional OFDM signal in SISO.

**Table 1 sensors-19-02696-t001:** ASP values with $B_{l}$.

$\left|ASP-ASP_{B^{l}/(B/16)=1}\right|/dB$	ASP/dB	$B_{l}/(B/M)$
0	−13.4	1
>16.6	<−30	6.77~7.25, 9.37~9.77

**Table 2 sensors-19-02696-t002:** CP values with Δk.

$\left|CP-CP_{\Delta k=0}\right|/dB$	CP/dB	|Δk/(B/MT)|
0	−19.5	1
>10.5	<−30	≥0.165

**Table 3 sensors-19-02696-t003:** Parameters of simulation signals.

Bandwidth		*B* = 400 MHz	
Subchirp duration		T=8 μs	
Number of subchirps		*M* = 16.	
Subchirp bandwidth of $s_{1}{}^{\prime }$	$B_{1}^{l}{}^{\prime }=6.8/16B$	Subchirp bandwidth of $s_{3}{}^{\prime }$	$B_{3}^{l}{}^{\prime }=9.4/16B$
Subchirp bandwidth of $s_{2}{}^{\prime }$	$B_{2}^{l}{}^{\prime }=7.09/16B$	Subchirp bandwidth of $s_{4}{}^{\prime }$	$B_{4}^{l}{}^{\prime }=9.77/16B$
Minimal interval of the chirp rates between each two signals		$\Delta k_{\min}$ = 0.29 $\left(\frac{B}{MT}\right)$	

**Table 4 sensors-19-02696-t004:** ASP and CP of previous and designed signals.

Waveforms(B = 400 M)	ASP/dB	CP/dB
Conventional OFDM (COFDM)	−13.4	−25
Li’s OFDM	−13.4	−26.3
Piecewise nonlinear frequency modulation OFDM (PNLFM-OFDM)	−25.0	−23.1
Segmented transmitting OFDM (STOFDM)	−30.2	−40.5

**Table 5 sensors-19-02696-t005:** ASP and CP of STOFDM signals with different number of signals.

Waveforms(B = 400 M)	ASP/dB	CP/dB
STOFDM($s_{1}{}^{\prime }$)	−30.2	\
STOFDM($s_{1}{}^{\prime }$ and $s_{2}{}^{\prime }$)	−30.2	−40.5
STOFDM($s_{1}{}^{\prime }$ and $s_{2}{}^{\prime }+s_{3}{}^{\prime }$)	−30.2	−35
STOFDM($s_{1}{}^{\prime }$ and $s_{2}{}^{\prime }+s_{3}{}^{\prime }+s_{4}{}^{\prime }$)	−30.2	−33.3
STOFDM($s_{1}{}^{\prime }+s_{2}{}^{\prime }$ and $s_{3}{}^{\prime }+s_{4}{}^{\prime }$)	−31.3	−37.1

**Table 6 sensors-19-02696-t006:** Range peak sidelobe ratios (RPSLR) of previous and designed STOFDM signals in single input single output (SISO) and MIMO radar.

Radar Model	Waveforms	RPSLR/dB
SISO	COFDM($s_{1}{}^{\prime }$)	−13.4
	Li’s OFDM	−13.4
	PNLFM-OFDM	−25.2
	STOFDM($s_{1}{}^{\prime }$)	−30.5
MIMO	STOFDM($s_{1}{}^{\prime }$ and $s_{2}{}^{\prime }$)	−28.7
	STOFDM($s_{1}{}^{\prime }$ and $s_{2}{}^{\prime }+s_{3}{}^{\prime }$)	−28.5
	STOFDM($s_{1}{}^{\prime }$ and $s_{2}{}^{\prime }+s_{3}{}^{\prime }+s_{4}{}^{\prime }$)	−27.3

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
