# Peer review of "OFDM Chirp Waveform Design Based on Subchirp Bandwidth Overlap and Segmented Transmitting for Low Correlation Interference in MIMO Radar"

_sensors, 2019, doi:10.3390/s19122696_

Reviewer 1 Report

The manuscript addresses the problem of design methods for the sub-chirp bandwidth, and transmit-receive mode is designed to reduce the autocorrelation sideline peaks in an OFDM chirp based MIMO radar. The authors also address the problem of reducing the cross-correlation peaks between two signals.

From a technical point of view, the manuscript seems correct. The system model and the approach developed by the authors is sound and to the Reviewer's knowledge original. Several plots support the results and prove the effectiveness of the approach. 

The only weak point in the technical part is the lack of a critical review of the drawbacks of the proposed solution. The authors seem to stress only the benefits, which of course have to be present, but some critical aspects should be highlighted more.

 The primary concern about this manuscript is the presentation, in particular, the language. Unfortunately, it was challenging for me to follow the presentation as many sentences (way too many) are grammatically wrong, and the whole language is far from being fluent. In this form, I hardly can recommend acceptance. So my suggestion is to encourage authors to seek the help of a native speaker or an expert to polish the language and the writing style of the whole manuscript. I believe that a higher quality level is necessary for this journal.

Author Response

Thanks a lot for the help. We are grateful to the editors and reviewers for their treasure time, great efforts and useful comments towards making this paper better in quality and readability. Now, after a careful revision, we present a point-by-point response to the comments and suggestions of the reviewers along with the new paper and hope it meets the requirement of publication. Below file are the replies to the comments.

Reviewer 2 Report

Overall, this paper represents an interesting contribution to MIMO-OFDM radar waveform design. However, before publication, it is my opinion that the authors should address the following major comments:

1)      The abstract should be rephrased so as to provide a gentler introduction to the considered problem.

2)      The following related works, dealing with  radar waveform design, should be discussed for completeness:

[R1] "Intrapulse radar-embedded communications via multiobjective optimization." IEEE Transactions on Aerospace and Electronic Systems 51.4 (2015): 2960-2974.

[R2] "OFDM MIMO radar with mutual-information waveform design for low-grazing angle tracking." IEEE Transactions on Signal Processing 58.6 (2010): 3152-3162.

[R3] "Robust waveform and filter bank design of polarimetric radar." IEEE Transactions on Aerospace and Electronic Systems 53.1 (2017): 370-384.

[R4] "Signal processing for passive radar using OFDM waveforms." IEEE Journal of Selected Topics in Signal Processing 4.1 (2010): 226-238.

[R5] "Pareto-theory for enabling covert intrapulse radar-embedded communications." 2015 IEEE Radar Conference (RadarCon). IEEE, 2015.

3)      Additionally, the discussion of related works should be rewritten to better highlight the current limitations of the existing works.

4)       Please rephrase the sentence “…dimensional information. To obtain its superiorities in range resolution and detection, the corresponding waveform should be designed with large enough bandwidth andlow ASP. And the CP between each two waveforms need to…” aiming at improved readability.

5)       Please avoid beginning sentences with “And….”, as well as contracted/informal writing, e.g. “What’s more”.

6)       Please add a notation paragraph at the end of Sec. I. 

7)       It would be useful for the generic reader adding a figure in Sec. II. depicting the considered system model. Additionally, Sec. II mostly lacks a clear statement of the considered problem.

8)       Please double-check the whole paper so as to avoid any kind of typo, e.g. “auto-corelation” and “cross-corelation” within some of the figures.

9)       Secs. 3 and 4 are a little bit hard to follow. In my opinion, they should be rephrased so as to provide a more streamlined and structured exposition. 

10)   In Sec. 5, the authors compare their waveform proposal only with simple LFM. At least a few waveform design baselines should be considered to provide a more solid comparative analysis.

11)   Conclusions should be enriched with what the authors consider to be further avenues of research.

Author Response

Thanks a lot for the help. We are grateful to the editors and reviewers for their treasure time, great efforts and useful comments towards making this paper better in quality and readability. Now, after a careful revision, we present a point-by-point response to the comments and suggestions of the reviewers along with the new paper and hope it meets the requirement of publication. Below file are the replies to the comments.

Round  2

Reviewer 1 Report

The reviewer is satisfied by the revised manuscript. The paper is now suitable for publication.

Reviewer 2 Report

Overall, this paper represents an interesting contribution to MIMO-OFDM radar waveform design. Additionally, the authors have satisfactorily addressed my previous comments and modified the manuscript accordingly.

Hence, I am glad to recommend the present paper for publication.